# A Metal-Containing NP Approach to Treat Methicillin-Resistant *Staphylococcus aureus* (MRSA): Prospects and Challenges

**DOI:** 10.3390/ma15175802

**Published:** 2022-08-23

**Authors:** Wendy Wai Yeng Yeo, Sathiya Maran, Amanda Shen-Yee Kong, Wan-Hee Cheng, Swee-Hua Erin Lim, Jiun-Yan Loh, Kok-Song Lai

**Affiliations:** 1School of Pharmacy, Monash University Malaysia, Jalan Lagoon Selatan, Bandar Sunway 47500, Malaysia; 2Faculty of Health and Life Sciences, INTI International University, Persiaran Perdana BBN, Putra Nilai, Nilai 71800, Malaysia; 3Health Sciences Division, Abu Dhabi Women’s College, Higher Colleges of Technology, Abu Dhabi 41012, United Arab Emirates; 4Centre of Research for Advanced Aquaculture (COORA), UCSI University, Cheras 56000, Malaysia

**Keywords:** NP, metals, *Staphylococcus aureus*, bacteria, antibacterial-resistance, biomedical applications

## Abstract

Methicillin-resistant *Staphylococcus aureus* (MRSA) is an important cause of pneumonia in humans, and it is associated with high morbidity and mortality rates, especially in immunocompromised patients. Its high rate of multidrug resistance led to an exploration of novel antimicrobials. Metal nanoparticles have shown potent antibacterial activity, thus instigating their application in MRSA. This review summarizes current insights of Metal-Containing NPs in treating MRSA. This review also provides an in-depth appraisal of opportunities and challenges in utilizing metal-NPs to treat MRSA.

## 1. Methicillin-Resistant *Staphylococcus aureus* (MRSA)

*Staphylococcus aureus* (Staph aureus or “Staph”) is a Gram-positive round-shaped bacterium which can often be found on the skin or nasal lining of an individual. *Staphylococcus aureus* is an opportunistic pathogen which causes severe infections such as sepsis, endocarditis, and pneumonia [1]. The overuse and inappropriate usage of antibiotics have led to the emergence of multi-drug-resistant pathogens such as methicillin-resistant *Staphylococcus aureus* (MRSA), a type of bacteria that is resistant to several widely used antibiotics is the leading cause of nosocomial and community infections. The rates of antimicrobial resistance (AMR) worldwide is increasing and poses a major public health threat [2,3].

The presence of MRSA strains can be identified via the genotypic method by detecting the highly conserved mecA gene that encodes penicillin-binding protein 2a, resulting in the resistance to methicillin, oxacillin, nafcillin, and cephalosporins [4]. Despite progress in preventing MRSA infections in health care settings, it still accounts for a significant amount of morbidity and mortality as well as a cost burden that is commonly associated with hospital-acquired infections.

## 2. Current Therapies in MRSA

Treatment options for MRSA include vancomycin, which remains as the initial treatment of drug-resistant Gram-positive infections for more than 60 years in addition to linezolid, daptomycin, telavancin or ceftaroline, depending on the clinical presentation [5,6,7]. In addition, the synergistic activity between vancomycin and several β-lactams against *S.aureus* as well as daptomycin-based therapy, ceftaroline-based therapy, linezolid-based therapy, quinupristin/dalfopristin, telavancin, trimethoprim/sulfamethoxazole-based therapy, and fosfomycin-based therapy has been widely studied to treat MRSA bacteremia [8,9].

These antibiotics work by inhibiting the DNA, RNA, protein and cell wall of the bacteria. However, antibiotic resistance is rising mainly due to over-prescription by health workers and over-usage by the public [10]. In addition to the improper stewardship of antimicrobial agents, the bacterium itself is also capable of overcoming the accumulation of antimicrobial agents inside the cells. It poses several resistances by limiting the uptake of a drug, inactivation of a drug, modification of a drug, overexpression of multidrug efflux pumps, and loss or mutation of porins [10,11]. Moreover, the adverse outcomes of infections by resistant bacteria are more detrimental compared to similar infections caused by susceptible strains. 

A antimicrobial photodynamic therapy (aPDT) has recently gained wide interest for its efficiency in killing various microorganisms, including MRSA [12]. The aPDT technique is based on the energy transferred to oxygen molecules which produces the radical species and hydrogen peroxide, resulting in necrosis or apoptosis of the bacteria. However, this therapy has yet to achieve efficacy or a cost-effective alternative option for MRSA treatment. In a recent study from Nasser and colleagues, they found that phages from the Siphoviride family and the Caudovirales order, which are isolated from sewage and stool samples, demonstrated effective bacterial lysis activity against clinical MRSA samples [13]. Thus, phage therapy could be another method to be employed as MRSA treatment as it is more economical and has less severe side effects on eukaryotic cells [14]. However, further studies are required to understand the mechanism of actions of phage therapy as there could be a possibility that phage therapy may develop the phage resistance phenomenon in the long run.

Naturally existing antimicrobial peptides exert protection against a variety of microorganisms, including bacteria, parasites, fungi, and viruses [15,16]. Nonetheless, the broad-spectrum of antimicrobial activity is not specific against MRSA and may pose hemolytic activity and unstable short half-life [17]. The limited treatment options available to overcome the multi-drug-resistant bacterial strains, this has posed a great challenge to the healthcare system that is associated with significant morbidity and mortality, especially in critically ill patients.

Therefore, current research is geared towards developing novel nanomaterials to overcome these challenges, as shown in Figure 1. Many studies are carried out to look into the field of nanomedicine, as this technology allows fine tuning of the sizes, morphologies, properties, and functionality of these nanomaterials for a more targeted and site-specific delivery of medicines. This has led to the exploration of nanomaterials with antimicrobial properties which is becoming an attractive alternative due to their nano size which can penetrate the cell membranes to fight against the bacteria [12,13,14]. Nanoparticles (NPs) which are significantly smaller than bacteria along with the large surface-area-to-volume ratio increase the contact area with target organisms and have become an alternative to combat MRSA and are been widely used for diagnostic or therapeutic purposes [15,16]. Table 1 summarizes findings from current studies on nanoparticles in treating MRSA.

## 3. Metal Nanoparticles

Metal NPs (NPs) are metals with a density of more than 5 g/cm^3^ [40]. Al (Al_2_O_3_), Au, Bi, Ce, Cu (CuI, CuO, and Cu_2_O), Fe (Fe_2_O_3_), Mg (MgO), Ti (TiO_2_), and Zn (ZnO) are metals that are fabricated to synthesize NPs [41,42,43,44] which are small in size and have a high surface-to-volume ratio, which impacts the antibacterial activity of NPs [40].

The ability to create reactive oxygen species (ROS) and their affinity to connect tightly with R-SH groups are the essential features that contribute to metal NPs’ bactericidal activity. Heavy metal ions of non-essential transition metals with high atomic numbers, such as Ag^+^ or Hg^2+^, can easily bind to SH groups, such as cysteine, disrupting enzyme function or breaking S–S bridges necessary to maintain the integrity of folded proteins, causing adverse effects on cell metabolism and physiology. 

Different bacterial killing mechanisms of metal NPs are employed such as inducing production of ROS, interaction with cellular membranes and structures (DNA, proteins), biomolecular damages, release of ions and ATP depletion [40,45,46]. Besides this, the nanopatterning and ease of modifying the surfaces at nano-scale may interrupt the adherence and colonization of bacteria that make NPs a promising approach against the resistance to traditional antibiotics [47]. As shown in Figure 2, there is a rising trend of research exploring the usage of nanomaterials for antibacterial resistance.

Among these nanomaterials, NPs are the most popularly studied inorganic NPs as they are notable for the non-specific bacterial toxicity mechanisms which broaden the antibacterial activity spectrum and increase the difficulties in bacterial resistance [46]. Examples of the NPs include silver NPs (AgNPs), copper NPs (CuNPs), and gold NPs (AuNPs). The metal NPs have emerged as promising alternatives to traditional antimicrobial antibiotics as NPs are postulated to target multiple biomolecules at a time and thus avoiding the development of resistant strains [40]. In addition, various factors such as formulation process, surrounding environment, bacterial own defense mechanisms as well as the physical characteristics (size, change, and charges) of the NPs also play a vital role in determining the effect on the antibacterial activities [40].

## 4. Synthesis of Metal Nanoparticles

The synthesis of metal NPs can be divided into two categories of top-down (destructive method) and bottom-up (constructive method) [48]. The top-down process involves physical and chemical approaches to break down bulk materials into Nano-sized particles. The bottom-up technique involves Nano-particle creation by the self-assembly of atoms, molecules, or clusters. In top-down approaches, externally controlled operations of cutting, milling, and shaping the materials into the appropriate sequence and shape are used. Physical synthesis involves Pyrolysis [49,50], nanolithography [51,52], thermolysis [53], and radiation-induced procedures [54,55]. However, this method has a significant drawback: the uneven surface structure of the produced metal NPs significantly impacts their physical and chemical properties [56].

Bottom-up approaches use chemical and biological procedures to synthesize NPs. The bottom-up strategy has been reported to be effective because it allows for significant control of the size, shape (physical parameters), and chemical makeup. This method is less expensive compared to the bottom-up method. Chemical [57,58], electrochemical [59,60,61], sonochemical [62], and green synthesis [63,64] are standard wet-chemical synthesis procedures used in the bottom-up approach. The disadvantage of this method is the purification of produced particles from their reaction mixture (toxic chemicals, organic solvents, and reagents) [48].

The exploitation of chemicals in the chemical synthesis of metal NPs is being questioned, as excessive use might lead to detrimental issues. This has resulted in the exploration environmentally friendly approaches such as green synthesis, biosystem synthesis, bacterial-based synthesis, fungus-based synthesis, algae-based synthesis, and plant-based synthesis [48,65]. These methods have shown to be viable alternatives to chemical NPs synthesis [66]. Over the last two decades, an exponential growth in publications utilising these methods has been observed, indicating the feasibility and safety of these methods [67,68,69,70,71,72,73].

The zeta potential (ZP) is widely used to characterize metal NPs in solution [74]. Purification and analysis of NPs are frequently focused on size and surface features which is the zeta potential [75]. Bacterial survival depends on their net surface charge, and changes in the surface charge can have physiological repercussions [76]. Antimicrobial compounds acting on bacterial surfaces have been studied for surface charge neutralisation as an antibacterial activity. Skoglund and colleagues (2017) studied how the physicochemical parameters of the solution, particle characteristics, and experimental settings affect measurements of the zeta potential of metal NPs in solutions of various characteristics [74]. This study further reported that in addition to reporting on mean values, zeta potentials should also be noted with intensity distribution curves to provide an accurate measurement.

A recent study by Hussein and colleagues (2021) reported that after conjugation of the negatively-charged *Punicagranatum* L. extract, the potential of chitosan-gold hybrid NPs changed from +53.1 6.7 mV to 31.0 6.0 mV, thus indicating a synergetic antibacterial effect against MRSA with MIC and MBC values of 15.6 and 62.5 g/mL, respectively [77]. Another study using AgNPs reported that the average zeta potentials of normal and autoclaved *S. aureus* MTCC 3160 were 50.2 and 3.2 mV [78]. Changes in zeta potential in bacterial surfaces can be linked to positively charged nanosilver and bacterial surface proteins. It is also reported that when bacterial cells were treated with ZnO NPs, the negative-charged bacteria interacted with positive-charged ZnO NPs due to electrostatic interaction, causing the charge to shift towards neutrality and causing the membrane permeability to change [79].

## 5. The Need for Metal-Containing Nanoparticles to Treat MRSA

Due to their nanosize and sharp size distribution, metal-containing NPs have been widely utilized in various fields, including environmental science, energy, food industry, catalysis, and medicine. Silver and copper NPs have been widely used as antifungal agents especially in controlling phytopathogenic fungi in agriculture [80]. Relating to the field of biomedical applications, the roles of metal-containing NPs are well exemplified, namely in antifungal activities, bioimaging, chemotherapy for cancerous cells, photothermal activities, as well as detection of glucose and hydrogen peroxide [81,82,83].

Besides this, metal-containing NPs also act as drug delivery carriers of various therapeutic agents such as antibodies, peptides, nucleic acids, and chemotherapy drugs [84]. In addition, metal-containing NPs also have been utilised as probes, namely in the bioimaging technique, which is used for disease diagnosis [85]. Moving towards cost-effective treatments and therapies, metal-containing NPs have gained attention for their tuneable size, shape, material, and surface in the healthcare sector. The gold and silver NPs which are reported to have antitumor properties are potential candidates for the treatment of breast, cervical, leukaemia treatments, and pancreatic cancers [86,87,88].

The potential usage of metal NPs against MRSA has gained high research interest such as silver NPs which have been long used as an antimicrobial agent and disinfectant for wound healing [25,89]. Interestingly, silver NPs have a broad spectrum of antibacterial, antifungal, and antiviral properties as well as anti-biofilm efficacy against MRSA [29,89,90,91]. The review of Aderibigbe et al. (2017) summarized the various formats of silver NPs, including (i) hexagonal and nanoplates silver NPs, (ii) combination of silver NPs with either cefazolin, mupirocin gentamycin, neomycin, tetracycline or vancomycin, and (iii) conjugation of cephalexin onto NPs was effective against *Staphylococcus aureus*.

Apart from utilizing metal NPs as antimicrobial agents for bacterial infection, these nanoscale materials together with engineered biological molecules such as enzymes, proteins, oligonucleotides, and polysaccharides have been explored in various applications, including therapy, diagnosis, bioimaging, biosensing, bioanalysis, biocatalysis, as well as cell and organ chips [92]. A metal NP has also been employed in surveilling antimicrobial resistance in patients. A recent study from Mohamed et al. (2021) showed a 90% clinical sensitivity and 95% clinical specificity in detecting antibiotic resistance in MRSA using patient swabs via the multicomponent nucleic acid enzyme−gold NP (MNAzyme-GNP) platform. This MNAzyme-GNP platform is also able to identify mecA resistance genes in uncultured nasal, groin, axilla, and wound swabs from patients with 90% clinical sensitivity and 95% clinical specificity.

Over the years, gold NPs are well-known for their chemically inert and biocompatible properties [93,94]. Moreover, their tunable physical characteristics (solubility, stability, and interaction with the environment) and the ease in conjugation with drugs and biomolecules have made gold particles an attractive choice for the bacteria infection treatment [25,95]. The synthetic flexibility of gold NPs in adjusting their size, shape, and surface properties and their strong absorption in the near-infrared region (NIR) make them an ideal candidate as a photothermal antimicrobial agent [39,40,96]. Upon exposure to 808 nm NIR laser, the protease-conjugated gold nanorods transform photon energy into heat, resulting in the disruption of *Staphylococcus aureus* bacteria membranes as these NPs can be activated under NIR irradiation [39]. Additionally, exotoxin clearance and biofilm removal were observed in this study which are important to address the issue of bacterial residues that persist in chronically ill patients.

Recently, metal NPs including Ag 10 nm, Ag 40 nm, Au 20 nm, and Pt 4 nm which are coated on 3D-printed biodegradable polymers have been widely utilized as medical supplies such as catheters, disposable materials, hospital bedding items as well as disposable antimicrobial linings and bandages [30]. These nanosized metal particles have played a major role in coating medical devices for their remarkable physical, chemical, and biological properties, especially silver NPs which can be used for the treatment of infections caused by highly antibiotic-resistant *Staphylococcus aureus* biofilms [31,32,91]. Likewise, Al-Taee et al. (2018) reported that gold NPs demonstrated a reduction of biofilm production and growth inhibition of MRSA isolated from clinical cases [95].

AgNPs are effective in altering the susceptibility of bacteria to antibiotic, thus stands as an effective antimicrobial agent. A recent study by Feizi and colleagues (2021) reported a significant decline in the growth of MRSA, indicating the credibility of AgNPs as a substantial substitute over conventional antibiotics in averting the biofilm-associated pathogenesis of MRSA [97]. The antibacterial activity of AgNPs was tested against five different strains of MRSA; MRSA1, MRSA2, MRSA3, MRSA4, and MRSA5, and research showed that AgNPs synthesized by wus1 had the most promising antibacterial activity with zones of inhibition of 15 mm, 15 mm, 14 mm, 18 mm, and 13 mm. On the other hand, AgNPs synthesized by *Penicillium* sp. showed a maximum zone of inhibition of 16 mm with 80 L of silver NPs [98].

## 6. Challenges in Using Metal-Containing NPs in Treating MRSA

One of the main challenges in using metal-containing NPs is clearance from the blood by the reticuloendothelial system (RES) in liver, spleen, and bone marrow, as they are not usually blood compatible [92,99] (Figure 3). The accumulation of metal-containing NPs may induce cytotoxicity in different cell types through apoptosis and necrosis [84,100,101,102] or even trigger coagulation response and, subsequently, activate a complementary cascade [92]. On the other hand, low retention of these metal-containing NPs may lead to low efficiency [102]. Thus, the examination of both short-term and long-term toxicity due to the cellular biodistribution and uptake of these metal-containing NPs should be taken into consideration during the development of effective NPs.

With the potential of metal-containing NPs in treating MRSA, there are still some challenges related with their long-term exposure from the aspect of NP clearance from our bodies. Elimination of NPs from the biological system is relatively low, leading towards prolonged accumulation in the system [81,102]. In addition, both the NPs and their degradation products can cause hemolysis, as they interrupt with the blood circulation which may pose a risk of organ dysfunction and damage. Therefore, selection of the charges for NPs are critical as negatively charged NPs are believed to have more cellular absorption as a result of plasma protein resistance by plasma proteins which eventually causes hemolysis and platelet aggregation [103].

It is essential to facilitate metal-containing NPs targeting specific cells in order to achieve the therapeutic response in treating MRSA. This is to reduce the loss of metal-containing NPs and NP amount and their activities in blood circulation besides minimizing damage to host cells and tissue [104]. Conjugation of existing antibiotics with highly targeted metal-containing NPs helps to selectively capture and kill MRSA by overloading defenses of drug-resistant bacteria [105]. A study from Wang and colleagues stated that target-oriented photo-functional NPs which are conjugated with both hematoporphyrin and monoclonal MRSA antibody killed selectively MRSA in L-929 cells [106]. This is in concordance with another study using platelet membrane-camouflaged NP PLT@Ag-MOF-Vanc (silver-containing NPs) that provides targeted drug delivery and reinforces the bactericidal effect on MRSA [107].

The safety of metal-containing NPs at the molecular level requires further investigation to better understand their potential toxicity. A recent study has reported that copper NPs cause severe consequences on the structure, function, stabilities, and activities of the metabolic enzymes (aldolase, catalase, lactate dehydrogenase, and quinone oxidoreductase) as compared to aluminum, ferum, nickel, and zinc NP [108]. This is rather worrying as NPs are capable of accessing the cytosol of our cells. Thus, it is of importance to explore the interactions between NPs and enzymes as enzymes are the fundamental biological catalysts responsible for all the biological regulation and metabolism in our bodies [109]. Several in vivo studies have shown that exposure of metal NPs decreased the activity of antioxidant enzymes in the brain [110,111,112].

At present, there are only a few clinical trials working on the effect of metal-containing NPs on nosocomial bacteria, such as the *Staphylococcus aureus* and *Pseudomonas aeruginosa*. This could be due to the high cost of using the high-throughput nanotechnology platform and equipment other than the scaled-up manufacturing of these nanomaterials [102]. Furthermore, inconsistency of size, morphology, and other properties of these metal-containing NPs for the large-scale production may contribute to the difficulties in producing efficient and potent NPs are also a major challenge [102,113].

Overall, metal-containing NPs have been extensively used not only in treating MRSA but also in various biomedical fields. However, there is still a limitation in the standardized studies on the various resistance mechanisms for common anti-MRSA antibiotics in *Staphylococcus aureus*, which includes (i) bacterial strains, (ii) fabrication and characterization of metal NPs, (iii) dosage, (iv) administration route, (v) clinical specimens, and others. As a result, the engineering of metal NPs to treat MRSA remains challenging to translate the findings from bench to bedside, as we need more comprehensive analysis to study the potential antibacterial mechanisms which govern the pharmacokinetics and pharmacodynamics of these metal NPs to define their therapeutic effect.

## 7. Prospects of Metal-Containing NPs in Treating MRSA

Silver NPs (AgNPs) are well-known for their broad-spectrum antibacterial properties against multidrug-resistant pathogens, especially in *S. aureus* [34,114]. It is also one of the most important NP, widely reported in biomedical application of wound healing, cell imaging, diagnosis, disease treatment, and contraceptive devices. Furthermore, the high surface of the AgNPs has been reported to increase the antibacterial capacity and bioavailability of the biomaterials [115]. Hamida and colleagues (2020) indicated that AgNPs have potential as an alternative antibacterial agent against MRSA, by targeting the virulence mechanism and biofilm formation, leading to bacterial death. Furthermore, introduction of AgNPs into protein apoferritin formed a stable Ag(I) complex that showed decomposition of MRSA in an in vitro assay [35].

Copper NPs (CuNPs) have also been shown to be effective in eradicating MRSA [46]. CuNPs releases Cu^2+^ ions which cause local pH and conductivity changes leading to disruption on bacterial cell membranes, thus altering the function of respiratory enzymes [104]. A recent study by Kannan and colleagues (2021) successfully synthesized CuNPs using chemical reduction and showed that CuNPs can potentially substitute conventional antibiotics in inhibiting biofilm-associated pathogenesis of MRSA.

Gold nano-particles (AuNPs) have been reported to revert MRSA resistance [42]. This study investigated attachment of amoxicillin to AuNPs in inhibiting clinical isolates and surmised an enhanced antibacterial efficacy. Another study by Kuo and colleagues (2019) showed that serum albumin-capped gold nanoclusters (BSA-AuNCs) have a great antibacterial activity against MRSA. AuNPs also show antibiotic resistance by hindering the high levels of β-lactamase produced by MRSA [33]. A recent study by Beha and colleagues (2021) utilizing multi-layer-coated gold NPs (MLGNPs) delivering antisense oligonucleotides (ASOs) showed silencing of MRSA.

## 8. Conclusions

MRSA infections poses a detrimental threat and solutions need to be considered. Antibiotics for treatment of MRSA are continuously misused and overprescribed, leading to uncontrollable bacterial resistance. It is evident in the literature that metal NPs exhibit antibacterial potency, holding a prominent role in MRSA. Besides these, recent trends concerning medicinal plants, natural drugs, synthetic chemical entities, bacteriophage therapies, and vaccines for MRSA are also being studied. Furthermore, extensive in vivo studies are peremptory in formulation, characterization, and testing metal NPs.

## Figures and Tables

**Figure 1 materials-15-05802-f001:**
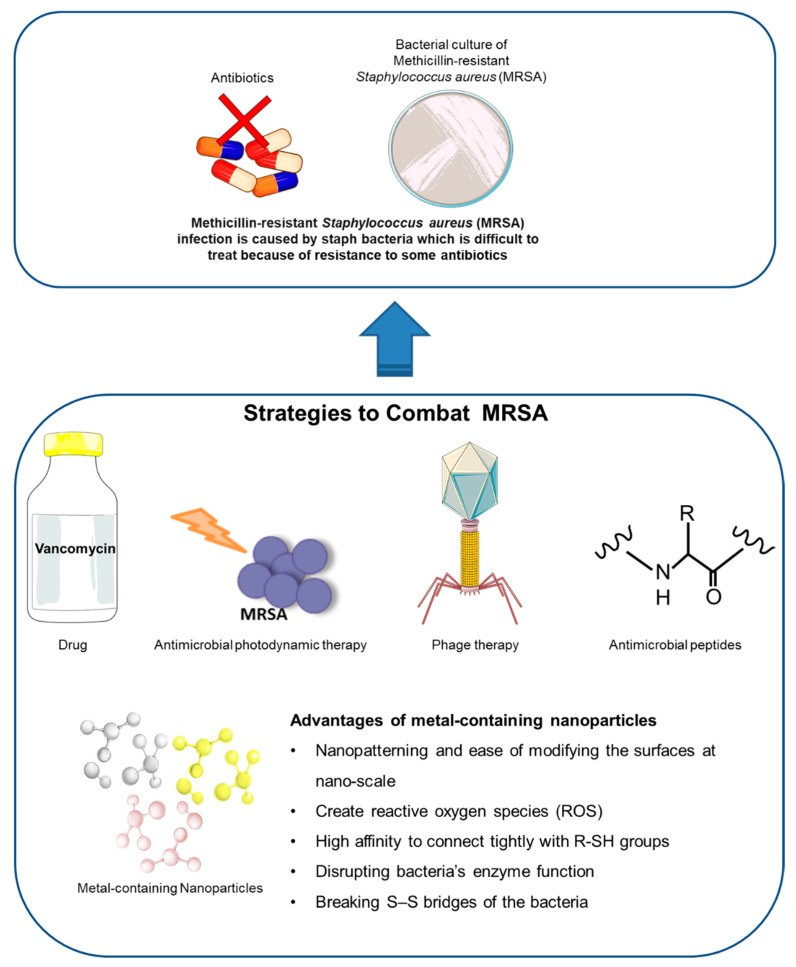
Emerging of metal-containing NP as an alternative strategy used to combat MRSA. Figure was modified using Servier Medical Art templates, which are licensed under a Creative Commons Attribution 3.0 Unported License (https://creativecommons.org/licenses/by/3.0/) [Accessed on 10 June 2022].

**Figure 2 materials-15-05802-f002:**
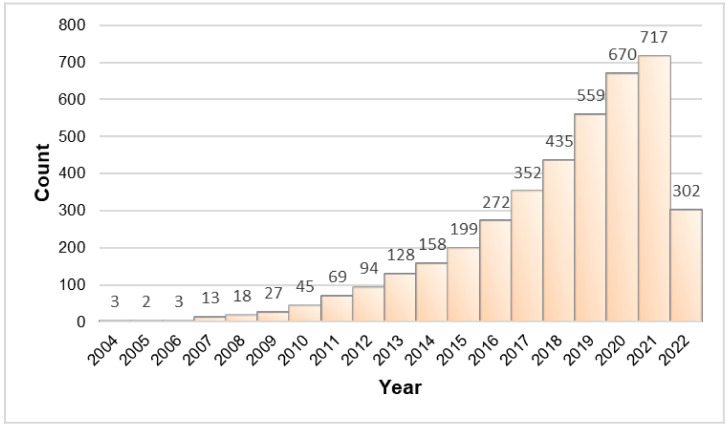
The increasing trend of research into nanomaterials and antibacterial resistance, as reflected by increasing publications in PubMed (updated: 7 June 2022).

**Figure 3 materials-15-05802-f003:**
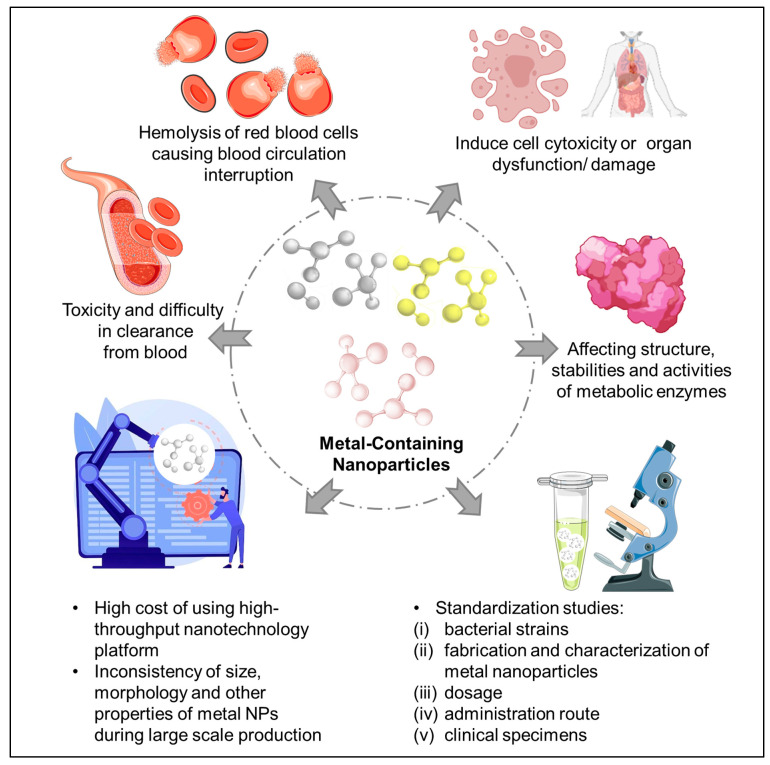
Current challenges of using metal NPs against MRSA for the translation to the clinics. Figure was modified using Servier Medical Art templates, which are licensed under a Creative Commons Attribution 3.0 Unported License (https://creativecommons.org/licenses/by/3.0/) [Accessed on 30 March 2022].

**Table 1 materials-15-05802-t001:** Summary of metal NP discussed in the present study.

Metal Nanoparticles	Findings	Reference
Silver	- Significant antagonistic action against MRSA with inhibition zones between 12 and 14 mm and minimum inhibitory concentration values between 1.56 and 12.5 g/mL were observed.	[18]
	- AgNPs were combined with commercial antibiotics resulted in increased antibacterial activity.	[19]
	- AgNps induced damage to MRSA biofilms with an increase in surface roughness.	[20]
	- AgNps characterized by smaller size showed higher antimicrobial activity compared to titanium dioxide and zinc oxide NPs. - The efficiency of the antimicrobial was strongly related to chemical composition, size, and concentration of NPs.	[21]
	- The most effective antibiofilm effect was seen on the coatings with the most Ag^+^ ion release, suggesting that Ag^+^ ions were responsible for the antibiofilm properties of nanosilver. - A positive correlation was observed between the Ag content of the coatings and biofilm found on the silicon substrate.	[22]
Biogenic Silver	- AgNps inhibited biofilm formation in MRSA. - AgNps showed an antimicrobial effect against MRSA by adhering to cell surface and penetrating into the bacterial cells thereby causing cell damage. - AgNps interacted with a bacterial membrane which resulted in reduced cellular respiration and induced lipid peroxidation. - Malondialdehyde was produced with a higher concentration of AgNps and longer incubation time, indicating the incresead free radical production in media.	[23]
	- Cell wall disruption and separation of plasma membrane from cell wall were observed in MRSA treated with biogenic AgNps.	[24]
Silver-containing, silica-based calcium phosphate	- Based on the results of microbial growth kinetics and colony-forming assay, Ag1/80S powders demonstrated an antibacterial effect against MRSA.	[25]
Apoferritin-Silver	- The introduction of AgNps into protein apoferritin formed a stable Ag(I) complex and reduced the growth depression of *S. aureus* culture.	[26]
Pexiganan and silver	- PLGA particles encapsulating the antimicrobial peptide pexiganan and embellished with Ag nanoparticles (Pex@NP-pTA-Ag) lessened antimicrobial infection.	[27]
Gold	- Inhibit adhesion and biofilm production of the tested bacterial strains.	[28]
	- Synergistic effect significantly diminished the drug resistance of MRSA by downregulating the expression of the drug-resistant gene mecA.	[29]
	- Upon exposure to 808 nm NIR laser, the protease-conjugated gold nanorods transformed photon energy into heat, resulting in disruption of *S. aureus* bacteria membranes. - This study also highlighted the activities of exotoxin clearance and biofilm removal with gold nanorods.	[30]
	- Gold nanorods coated by polymethacrylate with pendant carboxyl betaine groups (PCB-AuNRs) demonstrated better penetration and elimination of biofilms than non-surface charge transformable counterparts. - Upon NIR irradiation, PCB-AuNRs penetrated through the thickness of biofilm, indicating its excellent photothermal-induced killing effect.	[31]
	- No toxicity of AuNps was seen in mice and the antimicrobial effect of AuNps on MRSA was demonstrated. - AuNps inhibited the biofilm formation in MRSA. - A positive correlation was observed between the concentration of NPs and the inhibition zone of bacteria.	[32]
	- Attachment of amoxicillin to AuNps inhibited clinical isolates and enhanced antibacterial efficacy. - AuNps with amoxicillin demonstrated a clearance of MRSA infection in mice kidney and spleen which in turn increased the survival rate.	[33]
	- Bovine serum albumin-capped gold nanoclusters (BSA-AuNCs) demonstrated excellent antibacterial activity (70%–90%) against MRSA.	[34]
	- Multi-layer-coated gold nanoparticles (MLGNPs) delivering antisense oligonucleotides (ASOs) showed around 74% silencing of the *mecA* gene. - In the presence of oxacillin, the treatment of MLGNPs to MRSA demonstrated up to 71% of bacterial growth suppression, indicating the restoration of antibiotic susceptibility.	[35]
Multicomponent nucleic acid enzyme−gold	- MNAzyme-GNP platform revealed 90% clinical sensitivity and 95% clinical specificity in detecting antibiotic resistance in MRSA using patient swabs. - MNAzyme-GNP platform identified *mecA* resistance genes in uncultured nasal, groin, axilla, and wound swabs from patients with 90% clinical sensitivity and 95% clinical specificity.	[36]
	- Cu(II) release was measured using an Alizarin red assay after extended treatment with MRSA, demonstrating antibacterial effect.	[37]
Copper	- Copper oxide nanoparticles (CuO-Nps) showed an antimicrobial effect against MRSA. - CuO-Nps required higher concentrations to achieve a bactericidal effect as compared to CuNps and AgNps.	[38]
	- Liposomal synthesized CuNps demonstrated inhibition of biofilm, with cell damage and cell detachment seen upon treatment.	[39]

## Data Availability

Not applicable.

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
