# Peer review of "A Metal-Containing NP Approach to Treat Methicillin-Resistant Staphylococcus aureus (MRSA): Prospects and Challenges"

_materials, 2022, doi:10.3390/ma15175802_

Round 1

Reviewer 1 Report

In the proposed article "A Metal-Containing Nanoparticles Approach to treat Methicillin-resistant Staphylococcus aureus (MRSA): Prospects and Challenges" the authors presented the results of the literature analysis.

The manuscript might be interesting and relevant to the nanomaterials and, more in general, the nanomedicine community. However, it requires numerous changes before publication. There are some issues that could be additionally addressed.

1. The authors should revise the manuscript to provide a better description of the research state of the art and add more information on existing bactericidal agents (peptides, phages, and so on). I am aware that this is widely known in the community, but, in my opinion, it is necessary to briefly report this information in this manuscript to give a broader picture of the problem.

2. The authors should clarify information from Table1: "inhibition zones between 12 and 14 mm" is that wavelength?, or should it be written as mM for concentration? The authors should also add CuNP data.

3. The benefits of protein-CuNP and AgNP bioconjugates (nanobiohybrids, etc.)  should be considered.

4. In Fig.1 and 2: the resolution should be increased because the letters are too blurry.

5. The authors do not show the general picture. The primary purpose of the review is to create a new level of information based on existing pieces of the data and not to enumerate them.

Reviewer 2 Report

-The synthesis methods of metal nanoparticles and their effects not included. such as physical, chemical or biological method. or like green synthesis. These should be included in the manuscript.

The effect of size and zeta potential of metal nanoparticles on antibacterial activity has not been studied in detail.

In addition, although many metal particles such as Al, Mg, Se have antibacterial effects, they have never been mentioned. The existing metal nanoparticles in the literature have not been fully addressed. Metal NP that has no effect on MRSA should also be given. It should be shown comparatively if they have no effect. Table 1 should be rearranged. deficiencies must be corrected.

In my opinion, other properties of metal nanoparticles should also be briefly mentioned. (such as antiviral, antifungal, anticancer, or production of ROS) .

In table1 there is a sentence “ significant agonistic methicillin resistant (MRSA)….” . The abbreviation already includes " methicillin resistant ".

In the Manuscript, the names of bacteria are sometimes italic and sometimes not italic. Only one style should be preferred.

In fig 2, Ag, Au, Cu and others were named as metal nanoparticles. I guess the others are not named because they are not common. but what other metals it is makes me wonder. the metal type is very limited.

Not enough references were given for the review article. The review can be enriched by searching for new references on the subject. For example, doi.org/10.3390/c6030058     doi.org/10.1007/s00289-014-1235-x

In addition to metal nanoparticles, other trends may need to be mentioned when discussing antibiotic resistance.

Reviewer 3 Report

-          Line 88 add this ref., Scientific reports

-          Revise the whole manuscript for NPs instead of Nps

-          In Table 1: add this ref. 10.1016/j.biomaterials.2020.120344.

-          Lines 129, and 130: revise the writing of metal oxides e.g, Fe2O3 must be Fe2O3 and so on.,

-          It is better to include a high-resolution image for Figure 2

-          Line 172, what is the MNPs?

-          Lines 201-204: dose not male sense. Plz., try to rewrite to be more clear

-          Line 211: these zeta potential values are negative or positive?

·         Line 234: add these refs: Drug design, Development and Therapy 2021:15, 2035-2046.; doi: 10.1007/s11095-020-02938-1; https://doi.org/10.1080/03639045.2018.1483400

-          Line 238: add this ref: https://www.mdpi.com/2076-2607/10/7/1297#

-          Define abbreviations in line 251

-          Line 272: add this ref: https://www.mdpi.com/2076-2607/10/7/1297#

-          Line 283: which type of fungus used in this study

-          It is better to provide a high-resolution image for Fig.2

-          Line 357: Cu+2

-          It is better to include the different carrier systems for the metal nanoparticles against MRSA

-          Review should be revised in terms of abbreviations and be consistent all over the manuscript

Round 2

Reviewer 1 Report

The authors made all necessary changes. Now the article looks more solid.

Author Response

Dear reviewer,

Thank you for the kind feedback.

Reviewer 3 Report

After doing the corrections I did not see the included new references in the references list in the revised manuscript